# Policy Improvement using Language Feedback Models

**Victor Zhong***
University of Waterloo
Microsoft Research
`victor.zhong@uwaterloo.ca`

**Dipendra Misra**
Microsoft Research

**Xingdi Yuan**
Microsoft Research

**Marc-Alexandre Côté**
Microsoft Research

## Abstract

We introduce Language Feedback Models (LFMs) that identify desirable behaviour — actions that help achieve tasks specified in the instruction — for imitation learning in instruction following. To train LFMs, we obtain feedback from Large Language Models (LLMs) on visual trajectories verbalized to language descriptions. First, by using LFMs to identify desirable behaviour to imitate, we improve in task-completion rate over strong behavioural cloning baselines on three distinct language grounding environments (Touchdown, ScienceWorld, and ALFWorld). Second, imitation learning using LFMs outperform using LLMs as experts to directly predict actions, when controlling for the number of LLM output tokens. Third, LFMs generalize to unseen environments, improving task-completion rate by 3.5-12.0% through one round of adaptation. Finally, we modify LFMs to provide human-interpretable feedback without performance loss, allowing human verification of desirable behaviour for imitation learning.

## 1 Introduction

Sample-efficiency and generalizability are two primary challenges in learning instruction following agents in grounded environments [26, 23, 2]. First, we want an agent that is sample-efficient: it learns from few demonstrations of how to act according to instructions. Second, we want an agent that is generalizable: it should act successfully in novel environments according to new instructions after training. Reinforcement learning (RL; Sutton and Barto [40]) and imitation learning (IL; Schaal [32], Abbeel and Ng [1]) are two techniques for learning agents for instruction following in grounded environments. These techniques often require large numbers of trials and errors or expensive-to-obtain expert demonstrations. Recent work show that pretrained large language models (LLMs) exhibit sample-efficient learning through prompting and in-context learning for textual [10] and grounded problems such as robotic control [2]. However, for instruction following in grounded problems, current methods rely on LLMs on-line during inference, which is impractical and expensive.

We develop a sample-efficient and cost-effective technique that uses LLMs to train **Language Feedback Models** (**LFMs**) for policy improvement in instruction following. Figure 1 illustrates policy improvement using LFMs. Consider the task of interacting with objects in a kitchen to follow instructions shown in Figure 1(c). First, in Figure 1(a), given a grounded environment and a base policy (i.e. a behaviour cloned policy), we roll out the base policy to collect a small set of trajectories for different instructions. Next, we verbalize observations in the trajectory by describing scenes in language. For each instruction and verbalized trajectory pair, we query an LLM to provide feedback identifying which behaviour in the trajectory is productive to solving the task identified

---

*Corresponding author.

38th Conference on Neural Information Processing Systems (NeurIPS 2024).

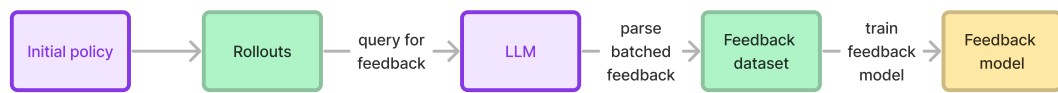

(a) Learning a small and cost-effective Language Feedback Model from LLM feedback. We roll out an initial policy, then prompt an LLM to provide feedback on what actions the policy took during the rollout were productive in achieving the task outlined in the instruction. We then use this data to train a feedback model that predicts whether an action is productive given the instruction.

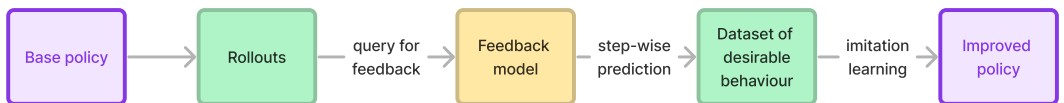

(b) Policy improvement by imitating desirable behaviour identified by a learned feedback model. Given the instruction, we roll out a base policy, then identify productive actions that help achieve tasks specified in the instruction using the trained feedback model. Finally, we update the base policy by imitating productive actions.

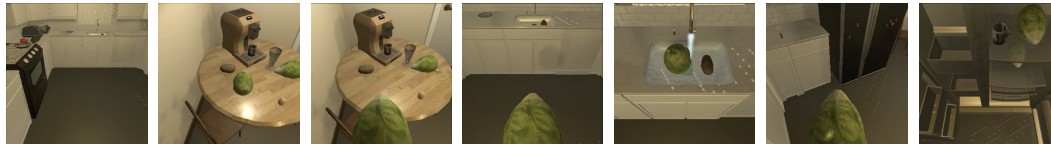

Instruction: clean some lettuce and put them in the fridge

(c) Example of desirable behaviour identified in ALFWorld, a kitchen instruction following benchmark.

Figure 1: Given an environment and instructions to follow, we assume a verbalization procedure that converts observations to language descriptions. Policy improvement using Language Feedback Model involves (a) training a feedback model, then (b) using it to identify desirable behaviour for policy improvement via imitation learning. The feedback model is yellow, other models purple, and generated intermediate data green. An example of LFM-identified behaviour is shown in (c).

in the instruction (i.e. answer yes or no). For instance, given an instruction "put a clean slice of lettuce in the refridgerator", GPT-4 [29] is able to deduce that key milestones are 1) find the lettuce, 2) slice it 3) wash it in the sink, and 4) put it in the fridge. Consequently, such an LLM is able to identify when an agent is exhibiting **desirable behaviour** conducive to solving tasks outlined in the instruction, for instance by taking the lettuce to the sink, versus undesirable behaviour, for instance by cooking the lettuce. We define desirable behaviour as productive actions that are constructive, task-beneficial, and effective in following the instruction. In other words, taking the action brings the agent closer (in terms of trajectory length) to accomplishing the task specified in the instruction. After collecting LLM feedback, we distill this world knowledge into a small and cost-effective LFM. Finally, in Figure 1(b), given a policy to improve on potentially new environments and instructions, we use the learned LFM to identify desirable actions on-line, then update the policy to imitate these actions. Crucially, this technique is sample-efficient in that it improves policy with no additional human-labeled demonstrations. Furthermore, this technique is cost-effective in that it requires few LLM interactions to collect an off-line dataset during LFM training (i.e. before deployment), as opposed to many LLM interactions on-line during policy improvement (i.e. after deployment).

Our findings are as follows: first, LFM policy improvement achieves consistent gains over strong behaviour cloned base policies on three grounded instruction following benchmarks in Touchdown [12], ScienceWorld [42], and ALFWorld [38]. Second, we compare LFMs against prompting LLMs to directly predict what actions to take, then imitating this LLM-predicted behaviour. On all benchmarks, using LFM feedback outperforms using LLMs as experts for imitation learning, given a fixed allocation of LLM output tokens. This gain is especially pronounced in environments with larger action spaces, such as ScienceWorld, where it is much easier to critique than to generate the correct action. Third, we show that learned feedback models generalize to unseen environments with new tasks and new transition functions. After training LFMs on training environments, we use them to identify desirable behaviour on test environments, which we then imitate to adapt the policy. A single round of adaptation achieves significant gains (3.5-12.0% task-completion) across all environments.

In addition to policy improvement, using LFM feedback offers two advantages over existing techniques such as using LLMs as expert policies for imitation learning. First, LFM improves policies on-line without additional expensive calls to LLMs. Second, LFM can offer human-interpretable feedback when identifying desirable behaviour to imitate. We show in Section 5.4 that LFMs can

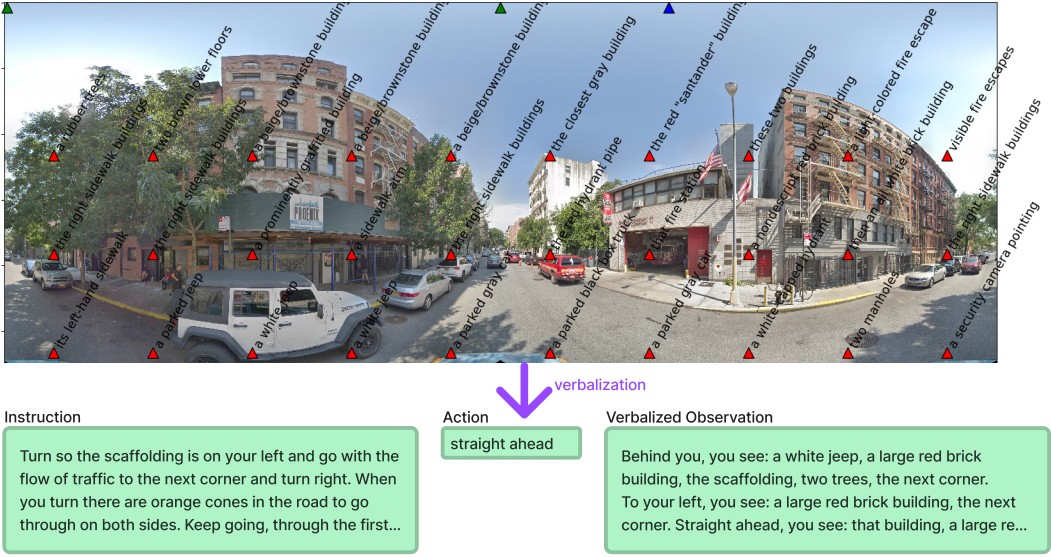

**Instruction**

Turn so the scaffolding is on your left and go with the flow of traffic to the next corner and turn right. When you turn there are orange cones in the road to go through on both sides. Keep going, through the first...

**Action**

straight ahead

**Verbalized Observation**

Behind you, you see: a white jeep, a large red brick building, the scaffolding, two trees, the next corner. To your left, you see: a large red brick building, the next corner. Straight ahead, you see: that building, a large re...

Figure 2: An example verbalization for Touchdown. We align CLIP image embeddings of panorama patches and language embeddings of common noun-phrases to populate a language template. Appendix D describes this procedure in detail. The blue arrow at the top indicate the agent's orientation while the green arrows indicate valid directions to proceed in.

be easily modified to provide not only desirable behaviour but why they were desirable, thereby allowing humans to inspect and validate imitation data used for policy improvement. Source code for our environments and experiments are available at github.com/vzhong/language_feedback_models. Videos of LFM feedback are available at language-feedback-models.github.io.

## 2 Background

**Language grounded instruction following.** In language-grounded instruction following, an agent is given an instruction $x$ specifying the task to achieve in the environment. Each turn, the agent receives a potentially partial observation $o_t$, and takes an action $a_t$ which causes the environment to transition to a new state. In the Figure 1(c) example, the agent observes a counter with objects such as a toaster, some lettuce, and a knife on top. To follow the instruction "put a clean slice of lettuce in the refridgerator", an effective agent may choose to grab a piece of lettuce. In the reinforcement learning setting, the environment additionally give the agent a reward after a desirable (positive reward) or undesirable (negative reward) action [40]. In this work, we consider long-horizon settings with only sparse and delayed task-completion rewards. Consequently, we focus on imitation learning from demonstrations as opposed to reinforcement learning from rewards [32].

**Online imitation learning.** In online imitation learning for instruction following, we are given an expert policy $\pi^*(a|x, o)$ and learn a policy $\pi_\theta(a|x, o)$ with parameters $\theta$. We first roll out the policy $\pi_\theta$. For each step $o_t^{(i)}$ of the rollout $\tau_i$, we optimize $\theta$ to imitate the action $a_t^{(i)}$ chosen by the expert $\pi^*(a|x, o_t^{(i)})$ when given the same observations: $\arg\min_\theta \mathbb{E}_{\pi^*} \left[ L\left(\pi_\theta(a|x, o_t^{(i)}), a_t^{(i)}\right)\right]$. Here, $L$ is step-wise cross-entropy between the policy's action distribution and the action chosen by the expert given the same observation: $L(*) = -\sum_{a' \in \mathcal{A}} \mathbb{1}\left[a' = a_t^{(i)}\right] \ln \pi_\theta(a = a' \mid x, o_t^{(i)})$.

**Behavioural cloning.** Online imitation learning assumes an expert policy that can be executed online to produce expert actions. For instance, given an expert, imitation learning assumes that this expert $\pi^*(a|x, o_t)$ provides corrective actions $a_t$ as the policy $\pi(a|x, o_t)$ runs. In many cases, this is impractical — a human-in-the-loop expert is expensive and inconvenient while an LLM expert is expensive and, as we show in our experiments, inaccurate. Alternatively, in behaviour cloning (BC), we instead collect an offline dataset of expert trajectories from which to clone expert behaviour [6, 41]. BC (or offline imitation learning) only asks the expert to perform the task $N$ times to collect $N$

trajectories $\{\tau_i\}_{i=1}^N$. Each $\tau_i$ consists of $M_i$ steps of observations and associated expert actions: $\tau_i = [o_1^{(i)}, a_1^{(i)}, \ldots, o_{M_i}^{(i)}, a_{M_i}^{(i)}]$ where $a_t^{(i)}$ is the action chosen by the expert $\pi^*(a|x, o_t^{(i)})$ given the observation $o_t^{(i)}$. We train policy $\pi_\theta$ to imitate the expert action, given the same observation seen by the expert, by minimizing the following objective: $\arg\min_\theta \frac{1}{N} \sum_i^N \frac{1}{M_i} \sum_t^{M_i} L\left(\pi_\theta(a|x, o_t^{(i)}), a_t^{(i)}\right)$. The key distinction between BC and imitation learning is that the former optimizes over trajectories under the expert policy while the latter optimizes over trajectories under the learned policy. Consequently, while BC is offline and easily batchable, it suffers from covariate shift/exposure bias [31, 7]. Like prior work in long-horizon instruction following in grounded environments [18, 12], we use BC to warm-start a strong base policy [4], which we then improve using imitation learning.

## 3 Language Feedback Model

How can we leverage world knowledge in LLMs to make policy learning more sample-efficient and generalizable? In this work, we use LLMs to distill a small and cost-effective Language Feedback Model to identify desirable behaviour from a base policy (Figure 1(a)). We then improve the base policy by imitating this desirable behaviour through batched imitation learning, without need for on-line LLMs (Figure 1(b)). Appendix E provides pseudo-code for the entire procedure for policy improvement using LFMs. A natural question is why not directly use LLMs as experts for action prediction. Section 5.4 shows that the using LLMs to learn feedback models results in higher policy improvement than using LLMs as experts for action prediction. Moreover, LFMs generalize to new environments unseen during training, thereby allowing policy improvement on new environments.

### 3.1 Verbalization

To leverage world knowledge in LLMs, we convert raw observations $o$ to language descriptions $v$ using a verbalization procedure $V$. Figure 2 illustrates verbalization for Touchdown [12], where the agent navigates Google Street View panorama images based on a given natural language instruction. First, we extract all noun-phrases (NPs) from instructions in the dataset and compute their CLIP language embedding. Given a visual observation, we compute CLIP visual embedding for each image patch, and align it to the NP with the highest cosine similarity between CLIP embeddings. We then combine aligned NPs with agent orientation to formulate an egocentric language description of the scene. This is described in more detail in Appendix D.

### 3.2 Learning a feedback model

**Naively learning from LLM feedback.** Given a verbalization procedure $V$, an instruction $x$, an LLM, and a policy $\pi_\theta$, we now describe a procedure to use the LLM's knowledge to improve $\pi_\theta$. First, we prompt the LLM to provide feedback on whether a particular action taken by the policy $\pi_\theta(a|x, v)$ is productive in achieving the tasks outlined in the instruction $x$. We then improve the policy $\pi_\theta$ by updating its parameters to imitate desirable behaviour determined by the LLM. Let : denote "such that". Let $\text{LLM}(x, v, a)$ return yes if and only if the LLM feedback indicates that action $a$ taken in verbalized state $v$ and instruction $x$ is productive. Given a set of instructions $X = \{x_i\}_1^N$, the optimization procedure is then $\arg\min_\theta \mathbb{E}_{v,a',x:\text{LLM}(x,v,a')=\text{yes}} L\left(\pi_\theta(a|x, v), a'\right)$. Here, instruction $x$ is sampled from $X$. Observations $v$ and actions $a'$ are sampled from rollouts of the policy $\pi_\theta$.

**Efficiently learning a language feedback model.** While the previously described naive learning is a reasonable procedure for using LLM feedback to improve the policy, it requires calling LLMs at each step during policy improvement. This is prohibitively expensive both in terms of query cost, because LLMs capable of giving desirable feedback are expensive to run, and training time, because generating feedback using large LLMs is slow. Instead of using the LLM at each step, we make a modification to collect LLM feedback over long horizons in batch [14] in order to train a small and cost-effective language feedback model.

First, for instructions $\{x^{(1)}, x^{(2)}, \ldots\}$ we roll out the base policy $\pi_\theta$ to collect a set of trajectories $\{\tau_1, \tau_2, \ldots\}$ consisting of verbalized observations and actions taken: $\tau_i = \{v_1^{(i)} \pi(x^{(i)}, v_1^{(i)}), v_2^{(i)} \pi(x^{(i)}, v_2^{(i)}), \ldots\}$. For each $\tau_i$, we prompt the LLM for feedback on which steps were productive in achieving the instruction $x^{(i)}$. Table 2's LFM row shows an example of

requesting feedback from GPT-4 on a rollout in ALFWorld, which is an instruction following benchmark in verbalized 3D kitchens. This LLM feedback is then parsed to identify the precise steps in which the base policy $\pi_\theta$ took a productive action towards achieving the goals outlined in the instruction. The set of desirable behaviour is compiled into a dataset $F$. Let $y^* = \text{LLM}(x, v, a)$ denote the feedback given by the LLM for the instructions $x$, observations $v$, and action $a$. We use the dataset $F = \{x^{(i)}, v, a, y^* \forall v, a \in \tau_i \forall x^{(i)}, \tau_i\}$ to train a small Language Feedback Model $f$:

$$\arg\min_\theta \sum_{(x,v,a,y^*)\in F} L\left(f_\theta\left(y \mid x, v, a\right), y^*\right). \tag{1}$$

Here, $L$ is the cross-entropy between the feedback model output $f_\theta$ and gold label $y^*$ from the LLM.

**Learning from language feedback.**   The naive learning procedure in Eq (3.2) updates the policy after each step using slow and expensive LLM feedback. Here, we instead update the policy in rounds using fast and cost-effective LFM feedback. In round $k$, we rollout the base policy $\pi^{(k)}$ and use the feedback model $f$ to collect a dataset $D_k$ of desirable behaviour. Let $a_t^{(k)}$ denote the action chosen by policy $\pi^{(k)}(a \mid x, v_t)$. Let $\text{DESIRABLE}(x, v, a) = f\left(y = \text{yes} \mid x, v, a\right) > f\left(y = \text{no} \mid x, v, a\right)$ return whether the feedback model predicts that action $a$ is desirable. In the $k$th round, we collect the dataset of desirable behaviour $D_k = \left\{\left(x, v_t, a_t^{(k)}\right) \forall t : \text{DESIRABLE}(x, v_t, a_t^{(k)})\right\}$, which we combine with previously collected behaviour to update the policy via imitation learning:

$$\theta^* = \arg\min_\theta \sum_{v_t, a_t \in \cup_{i=1}^k D_i} L\left(\pi^{(k)}(a \mid x, v_t), a_t\right). \tag{2}$$

In the next round, we set the base policy $\pi^{(k+1)}$ parameters to $\theta^*$. Should demonstrations be available, we initialize the base policy at $k = 1$ to the BC policy, and train on both demonstrations and identified desirable behaviour during subsequent rounds (i.e. $\cup_{i=0}^k D_i$ where $D_0$ are demos used to train BC).

## 4   Related Work

**Instruction following in grounded environments.**   Instruction following in grounded environments has been explored in settings such as navigation [11, 18, 12], game-playing [3, 48], and robotics [8, 37, 9]. However, most prior work model environment observations separately from language instructions by using specialized encoders (e.g. RESNET [19], BERT [16], CLIP [30]), then learn from data how to associate raw observations with language instructions. Instead of solely using raw observations, more recent work verbalize raw observations to describe environments in language [38, 49, 34]. In doing so, observations and instructions can be directly jointly reasoned over using language models to achieve more efficient and generalizable learning through large-scale pretraining. We build on this last direction by verbalizing raw observations into language descriptions to train language policies. However, unlike prior work that train language models to predict next actions, we develop language feedback models that critique verbalized observations and behaviour.

**LLM agents in language settings.**   LLMs exhibit reasoning abilities after pretraining on vast quantities of text [10, 43]. A number of recent work on LLMs language agents exploit this reasoning ability. Nakano et al. [28], Yao et al. [46] Deng et al. [15] train instruction following language agents to interact with web browsers to answer questions or interact with web pages. Ahn et al. [2] show that a language agent can be connected with verbalized robots via API interfaces for robotic control. Xie et al. [44] use large visual language models (VLMs) for instruction following in virtual machines, but show that LLMs with verbalization outperform VLMs. While powerful, these prior work are limited in that they require querying an expensive LLM on-line. In contrast, our work examines settings where an LLM is not available on-line. Specially, we use LLMs to collect a small set of off-line data for training LFMs. The small and cost-effective LFMs are then used to identified desirable behaviour for on-line policy improvement without additional interactions with the LLM.

**Learning from feedback.**   Recent work enhance language agents by augmenting them with feedback. Ziegler et al. [50], Stiennon et al. [39], and Bai et al. [5] learn reward models from human preference to improve policies via reinforcement learning (RL). Instead of using human feedback, Bai et al. [5] and Lee et al. [24] use LLM feedback to train a separate reward model for RL for textual

Table 1: Examples of verbalization. We abbreviate long verbalized observations using "...".

| Benchmark | Context | Action |
|---|---|---|
| ALFWorld | Task: heat some egg and put it in diningtable.
Observation: You arrive at loc 12. On the sinkbasin 1, you see...
T-1 Observation: You are in the middle... Action: go to sinkbasin 1
T-2 Observation: ... | go to
microwave 1 |
| ScienceWorld | Task: Your task is to find a(n) living thing. First, focus on the thing. Then,
move it to the purple box in the bathroom.
Observation: You move to the kitchen. This room is called the kitchen. In it,
you see: | the agent | a substance called air | a chair. On the chair is...
In your inventory, you see: | an orange...
T-1 Observation: The door is now open. Action: go to kitchen
T-2 Observation... Action: open door to kitchen | open door
to outside |
| Touchdown | Task: Follow the flow of traffic, with the row of flowers on your left and
make a left at the intersection. There will be a white billboard...
Observation: behind you, you see: the right lane intersection, a large...
T-1 Observation: behind you, slightly... Action: slightly to your left ... | straight
ahead |

alignment. Huang et al. [21], Yao et al. [46], and Shinn et al. [35] use LLMs to reason about potential resolutions to failed actions. Yuan et al. [47] use LLMs to generate new prompts and corresponding responses, then use an LLM reward model to identify good prompt-response pairs for self-improvement in text generation alignment. Unlike these approaches, we do not use LLMs during on-line policy improvement. We train an initial small language feedback model from offline LLM data, then use this small feedback model for policy improvement. Additionally, we focus on-line improvement via language feedback for long-horizon, sparse reward, grounded environments instead of text generation alignment. Our procedure for batched, on-line imitation learning is similar to DAGGER [31], which we compare to in Appendix F. However, we collect batched expert feedback to identify desirable behaviour instead of corrective actions. Klissarov et al. [22] and Du et al. [17] are recent works that describe learning from feedback approaches complementary to ours. The former learns preference models based on pairwise observations while the latter uses LLMs to suggest exploratory goals during training. Unlike these works, which assume that the underlying goal is the same between training and inference, we consider settings where training and evaluation goals are different. That said, one can expand these approaches to generalize to unseen environments by adapting a preference model during inference [22] and by goal-conditioned subgoal generation during inference [17]. However, unlike LFM, these modifications would then rely on calling LLMs during inference.

## 5   Experiments and Analysis

We evaluate using LFM s for policy improvement on three distinct language grounding benchmarks. Formally, the **environment**s from a **benchmark** are distinct partially-observed Markov Decision Processes that share some (or all) of the environment dynamics but have different instructions, observations, and/or action space.

### 5.1   Evaluation benchmarks

Table 1 shows examples of verbalized environments and tasks from each benchmark. Each benchmark provides distinct training and test environments to test generalization. In each environment, the agent takes actions to perform tasks outlined in a language instruction. The task is considered completed if and only if the agent solves the tasks within the preallocated number of steps. We evaluate using task-completion rate over test environments. The statistics from each benchmark is shown in Appendix D Table 6. These three benchmarks share challenges in sparse, delayed reward, partial observability, and compositional generalization to unseen tasks and environments.

**ALFWorld** is a verbalization of ALFRED [36], a natural language instruction following benchmark set in a 3D simulated kitchen. Here, the agent interacts with objects in kitchens to achieve compositional goals such as cleaning then microwaving potatoes. In ALFWorld [38], raw state information from ALFRED are used to populate language templates that describe observations in language.

Table 2: LLM prompts used to collect desirable behaviour. ACTPRED uses LLMs to directly generate actions for each step, whereas LFM uses LLMs to generate batch feedback that identify which taken actions were productive. For brevity, we abbreviate long verbalized observations using "...". "Before" contains the observation before the first step in the batch.

| ACTPRED | |
|---|---|
| **Prompt** | Your task is: look at alarmclock under the desklamp. |
| | You see: you are in the middle of a room. looking quickly around you, you see a bed 1... |
| | what do you decide to do? available actions: examine shelf 1, examine shelf 2, go to bed... |
| | You decide to: go to desk 1. |
| | You see: you arrive at desk 1. what do you decide to do? available actions: examine desk... |
| | You decide to: |
| **LLM Output** | examine desk 1 |
| **LFM** | |
| **Prompt** | You will be shown a playthrough for solving a task. |
| | Task: put two candle in drawer. |
| | Before: You open the drawer 6. The drawer 6 is open. In it, you see nothing. |
| | Step 27. Your action: close drawer 6. Result: You close the drawer 6... |
| | Step 28. Your action: put candle 3 in/on drawer 1. Result: You put the candle 3 in... |
| | Is the player on the right track to solve the task? |
| | Answer yes or no. If yes, list the helpful steps by the step number in bullet form. |
| **LLM Output** | Yes |
| | - Step 28 |
| | - Step 29... |

**ScienceWorld** is a textual simulation benchmark for basic science experiments [42]. The agent interacts with objects to conduct experiments specified in natural language, such as determining the boiling temperature of a material. ScienceWorld is uniquely challenging to due the large amount of variations in task types (30), and parametric variations (10-1400) such as the specific substance to be melted. Furthermore, ScienceWorld has a substantially larger action space and longer horizon tasks.

**Touchdown** is a navigation benchmark where the agent navigates Google Street View images to follow long, compositional instructions [12]. Touchdown requires jointly reasoning over natural images from Google Streetview with occlusion and multi-sentence natural language instructions that describe long-horizon goals. We introduce a new verbalization procedure for Touchdown based on matching noun-phrases and image patches with CLIP embeddings to populate egocentric language templates. Behaviour cloning using our verbalization is detailed in Appendix D. Touchdown considers multiple subtasks, in this work we only test the agent's ability to arrive at the correct location according to the instruction.

## 5.2 Methods

We train BC baseline policies using existing demonstrations. We examine three different techniques for improving the BC policy. Table 2 shows examples of LLM prompts used for each technique.

**ACTPRED: imitation learning from LLM experts.** We compare to directly using LLMs as experts to predict actions for imitation learning. First, we execute $k$ steps of the base policy, then query the LLM for the next action $a$ given the instruction $x$ and the verbalized observations $v$. We repeatedly collect examples $(x, v, a)$, then train the policy using this collected data and BC demonstrations.

**LFM: imitation learning using feedback models.** We learn a small and cost-effective feedback model described in Section 3.2 to identify desirable behaviour for imitation learning. First, we learn a feedback model on the training environments. Second, we use the feedback model to identify desirable behaviour in the training environments for policy improvement via imitation learning. To collect LLM feedback for training LFMs, we collect one rollout for each environment in a benchmark and sample 10k 20-step windows from the rollouts. Crucially, we limit the amount of feedback data collected from the LLM such that the number of output tokens produced by the LLM is identical to ACTPRED (we use 100k GPT-2 tokens for all benchmarks). For LFM, we collect feedback for as many windows as possible until we exceed 100k output tokens, then use this feedback to train the

LFM. For ACTPRED, we label actions until we exceed 100k output tokens, then combine this labeled set with demonstrations to train the policy. This limitation on LLM interactions answers whether the feedback model is more cost-effective than direct action prediction for imitation learning.

**LFMA: adaptation using feedback models.** LFM only imitates desirable behaviour in training environments. In contrast, LFMA adapts the policy to test environments. Given new test environments, we identify desirable behaviour using feedback models trained on the training environments, then perform one round of imitation learning to adapt to new test environments. This experiment tests whether language feedback models generalize to new environments, and whether we can use their feedback to adapt policies to new environments without using LLMs nor additional demonstrations.

## 5.3 Experiment details

We use the GPT-4 (2023-03-15) for action prediction and feedback, and finetune 770M FLAN-T5 [13] for policy and feedback models. Verbalized observations $v$ contain the most recent 20 steps. We train models for 10k steps with batch 20, learning rate 5e-5, and early stopping over validation demos. For ACTPRED and LFM, we limit the amount of LLM usage to 100k GPT-2 tokens. Touchdown verbalization uses `vit-large-patch14`. Appendix H details GPU usage.

**Feedback model training and inference.** We train LFMs using LLM feedback over 20-step windows. We then parse feedback to identify whether the action taken in each step was productive to solving the tasks outlined in the instructions. We subsample the feedback data to obtain an even split of productive and not-productive actions. This data is split into a 80% train/20% validation dataset to train the LFM.

Table 3: Task completion rates of behaviour cloning BC, imitation learning (IL) using LLM expert ACTPRED, and IL using LFM. On held-out test environments, LFM outperforms other methods on all benchmarks. ACTPRED and LFM are limited to 100k output tokens of GPT-4 interactions. Further adaptation to the new environments using LFM results in significant additional gains (LFMA). Errors are standard deviations across 3 seeds. Previous SOTA are Micheli and Fleuret [27] for ALFWorld, Lin et al. [25] for ScienceWorld, and Schumann and Riezler [33] for Touchdown. Unlike Lin et al. [25], our methods do not use ScienceWorld-specific custom room tracking nor action reranking.

|  | ALF | SciWorld | TD |
|---|---|---|---|
| Prev SOTA | 57.6 | 45.8 | 29.3 |
| GPT-4 zeroshot | 3.0 | 1.3 | 3.2 |
| BC | $62.6 \pm 0.4$ | $45.8 \pm 0.6$ | $57.5 \pm 0.3$ |
| ACTPRED | $56.0 \pm 0.7$ | $39.0 \pm 0.7$ | $58.0 \pm 0.4$ |
| LFM | $\mathbf{64.1} \pm 0.3$ | $\mathbf{47.1} \pm 0.5$ | $\mathbf{59.7} \pm 0.4$ |
| LFMA 1 rnd | $74.6 \pm 1.1$ | $49.3 \pm 0.9$ | $62.8 \pm 1.1$ |
| LFMA 2 rnds | $76.5 \pm 1.3$ | $50.4 \pm 1.0$ | $63.5 \pm 1.2$ |

**Policy training and inference.** To train policies, we fine-tune language models to minimize token-wise cross-entropy of the ground-truth verbalized action. During inference, we consider a (potentially very large) set of plausible actions given by the environment. For each action, we evaluate the policy's language model perplexity, and choose the action with the minimum perplexity averaged over tokens.

## 5.4 Results and discussion

Table 3 shows the performance of the policy behaviour cloned from demonstrations BC, imitation learned from LLMs using action prediction ACTPRED, and imitation learned from LFM. For LFMs, we show zero-shot (LFM) as well as adaptation (LFMA) results.

**LFMs improves policy performance across all benchmarks.** Table 3 shows that LFM improves upon the strong behaviour cloning baseline policy BC in all benchmarks. Table 5 shows examples of LFM-identified desirable behaviour. This shows that LFMs are an effective means to leverage the knowledge in pretrained LLMs for policy improvement in language-grounded environments, which agree with human-identified desirable behaviour. Appendix G also compares GPT-4 to the open-source LLAMA 2 70B for training feedback models using human evaluation. We find that GPT-4 consistently outperforms LLAMA 2, which tends to identify spurious desirable behaviour.

**Learning LFMs is more cost-effective than using LLMs for action prediction.** Assuming the same LLM output-token quota, Table 3 compares using LLMs to train feedback models (LFM) to using LLMs to predict actions (ACTPRED) for policy improvement. Specifically, ACTPRED tends

Table 5: Example of detailed human-interpretable feedback. The prompt and output differences between succinct feedback (Table 2) and detailed feedback are highlighted in red. Note that the second row shows shows batched LLM inference, which is used to train a step-wise feedback model. The step-wise inference of the feedback model is shown in the last row. We leave providing summaries and suggestions with the learned feedback model to future work.

**Prompt**
You will be shown a playthrough for solving a task. Task: put two candle in drawer. Before: You open the drawer 6. The drawer 6 is... Step 21. Your action: close drawer 6. Result: You ... Is the player on the right track to solve the task?
Next under heading #Summary, summarize in one sentence what the player doing at a high level. Finally under heading #Improvement, describe how the player can improve their strategy to solve the task.

**LLM output**
Yes
- Step 28: The player locates the candles by arriving at countertop 1, where a candle is available.
- Step 29: The player successfully takes candle 3 from countertop 1, demonstrating the ability to collect...
#Summary
The player is gathering candles and placing them in a drawer, working towards completing the task of...
#Improvement
The player can improve their strategy by: not closing drawers unnecessarily...

**Learned feedback model output**
Yes. The player successfully locates the candle...
Yes. The player correctly takes the candle from...

to predict spurious actions, especially for complex environments with large actions spaces such as ScienceWorld. In contrast, the difficulty in identifying productive actions is independent of the action space, and LFM consistently improves policy even with large action spaces. This shows that LFMs is a more cost-effective means use LLMs for policy improvement compared to using LLMs as expert policies for imitation learning.

**LFMs generalize to new environments, allowing for policy adaptation without additional LLM usage nor demonstrations.** Table 4 shows that LFMs trained during language feedback learning can accurately recognize desirable behaviour in new environments. Table 3 shows that imitating this behaviour obtains significant policy improvement across all benchmarks. This shows that LFMs generalize to new environments, which allows for policy adaptation to new environments despite not having demonstrations nor LLM access. Appendix I shows additional experiments that demonstrate correlation between LLM/LFM feedback and true state values.

**LFMs can provide human-interpretable feedback, allowing human-in-the-loop verification during policy improvement.** LFMs improve policy performance with succinct feedback. Here, we extend them to additionally provide detailed explanations. Consider an instruction "turn left when you see the stop sign then go to the second building on the right". Suppose that in the current step the agent proceeds straight, arriving at the stop sign. Instead of a feedback saying "yes" (i.e. the action was productive), the LFM can provide a human-interpretable explanation for why this action was productive (i.e. "yes because you found the stop sign where you are supposed to turn"). Table 5 shows that we can enhance LFM to produce detailed feedback by training detailed feedback prompted from LLMs. Specifically, we train a detailed LFMD to simultaneously identify productive actions, summarize agent intent, and suggest potential high level recovery strategies.

Table 4: Feedback performance of LFM. We measure F1 score of the productive/not-productive predictions made by the learned LFM using the LLM predictions as ground truth. We observe no significant performance degradation when using a much more detailed feedback model (LFMD) that also provides explanations behind the feedback, summaries of agent behaviour, and strategy suggestions.

|       | ALF  | SciWorld | TD   |
|-------|------|----------|------|
| LFM   | 93.2 | 83.7     | 43.9 |
| LFMD  | 92.0 | 82.5     | 42.5 |

Table 4 shows that surprisingly, LFMD that produce detailed feedback perform similarly to those that provide succinct feedback. This shows that Language Feedback Models can be used to provide accurate feedback interpretable to humans. While interpretable feedback requires more costly LLM usage, it allow for human-in-the loop verification of desirable behaviour identified by the LFM. Consequently, interpretable LFMs promotes user trust in the quality of the imitation learning data and subsequent policy behaviour.

# 6 Conclusion

We introduced Language Feedback Models that identify desirable behaviour for imitation learning. On three instruction following benchmarks, small and cost-effective LFMs consistently outperform BC baselines and using LLMs as experts for imitation learning, without using LLMs during policy improvement. In addition, LFMs generalize and provide significant policy adaptation gains on new environments, without using LLMs nor new demonstrations. Finally, LFMs, can provide detailed human-interpretable feedback that human verification of imitation data. We advocate for future exploration of how to exploit detailed LFMs, such as learning dense, subgoal-aware reward models for RL, and trustworthy policies with human verification.

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

# A    Limitations

This work proposes using LLM feedback to improve policies for long-horizon planning in grounded environments. It assumes access to a verbalization module that faithfully describes observations in language. For some practical problems (e.g. rich scenes in operating systems Xie et al. [45]), verbalization has been shown to be a difficult problem, which limits the capability of language feedback models. Moreover, LLMs have been shown to hallucinate, especially in grounded settings that are uncharacteristic in internet pretraining data [42]. Consequently, LLMs may provide inaccurate feedback, which limits policy improvement gains.

Our empirical results demonstrate consistent policy improvement using LFMs on three distinct language grounding benchmarks. These benchmarks cover kitchen interactions (AFLWorld), scientific experiments (ScienceWorld), and real-scene navigation (Touchdown). Future work should investigate the application of LLM feedback learning to real-world robotics, as an important test-bed for LLM ability to provide precise feedback on real-world observations.

While this work significantly reduces the computational burden of using language models as policies in grounded environments by extracting world knowledge from LLMs into small LMs using language feedback, the LM policies we use still require large GPUs to train. Specifically, we note significant performance degradation (e.g. 10% task completion) when changing from 770M Flan-T5 to the smaller 250M variant. Future work should investigate techniques to further reduce model size so that LMs can be tractably used as policies on small devices such as phones.

# B    Broader Impacts

Although this work develops LM as policies in grounded environments, the outputs these policies generate are specific (i.e. plans) to the benchmark environments used. Consequently, the generated outputs are not suitable for malicious use (e.g. disinformation, fake profiles). One potential misuse of our proposed method lies in a central party providing a malicious feedback model, which provides negative reinforcement to good behaviour and positive reinforcement to bad behaviour. Learning from such a feedback model can potentially result in malicious downstream policies.

# C    Licenses for existing assets

We will release our code and resources under the MIT license. This work uses assets from the Flan-T5 series of models, OpenAI GPT-4, Llama-2, and the three benchmarks. The licenses are as follows:

- Flan-T5: Apache 2
- OpenAI GPT-4: fair use policy
- Llama-2: community license
- ALFWorld: MIT
- ScienceWorld: Apache 2
- Touchdown: Attribution 4 International

Table 6: Statistics from benchmarks as measured by training demonstrations. The are the average number of GPT-2 tokens in the instruction, verbalized observation, and action; the average demonstration steps; the average number of plausible actions in a state; the number of unique actions, instructions, and observations; and finally the number of training demonstrations.

|  | ALFWorld | SciWorld | Touchdown |
|---|---|---|---|
| Ins len $\|x\|$ | 8.8 | 64.7 | 93.4 |
| Obs len $\|v\|$ | 23.9 | 239.4 | 284.9 |
| Act len $\|a\|$ | 4.5 | 6.0 | 2.4 |
| Traj len $\|\tau\|$ | 19.7 | 55.1 | 34.2 |
| \|Act space\| | 29.9 | 1.9k | 2.1 |
| # act $\|\{a\}\|$ | 2.6k | 2.4k | 8 |
| # ins $\|\{\tau\}\|$ | 1.0k | 1.2k | 6.5k |
| # obs $\|\{v\}\|$ | 18.2k | 157k | 34.3k |
| # demos | 3.5k | 3.6k | 6.5k |

# D  Verbalization of visual environments

How can we leverage the world knowledge learned by LLMs from pretraining on vast quantities of text? In many instruction following problems, environment observations are inherently visual. In this section, we describe a verbalization procedure that converts visual observations to language descriptions, so that LLMs can make inferences by jointly referring to the instruction and environment observations. Specifically, we use Touchdown as an example.

As shown in Figure 2, Touchdown [11] is primarily a visual navigation problem. Given a set of connected Google Streetview panoramas that represent neighbourhoods in Manhattan, an agent must follow long, complex natural language instructions to navigate to the correct location. Crucial to this task of navigation are **landmarks** referred to by the instructions. For instance, the instruction "turn so the **scaffolding** is on your left and... to the **next corner** and turn right..." refers to the landmarks **scaffolding** and **next corner**. Prior work in verbalization use LLMs to identify landmarks [34]. In this work, we take the set of common noun-phrases in the corpus of instructions to be landmarks.

**Extracting aligned noun-phrase annotations for visual patches**  First, we identify all noun-phrases using SPACY [20]. Given a visual scene, we divide the scene into 300x300 pixel non-overlapping patches. For each patch, we identify the noun-phrase with the highest cosine similarity between the noun-phrase text embedding and the image patch embedding. We use text and visual encoders from CLIP [30] to extract embeddings for each modality. For patches with no aligned noun-phrase with cosine similarity greater than 0.2, we do not provide annotated a noun-phrase.

**Converting to verbalized observations**  To obtain verbalized observations in the form of an egocentric scene description, we consider the direction the agent is facing (shown in blue) as well the directions of possible next steps (shown in green). The noun-phrases identified in the scene are then categorized into 8 directions in 45-degree increments, relative to the agent's current orientation: straight ahead (337.5 to 22.5), slightly to your right (22.5 to 67.5), to your left (67.5 to 112.5), behind you, slightly to your right (112.5 to 157.5), behind you (157.5 to 202.5), behind you, slightly to your left (202.5 to 247.5), to your left (247.5 to 292.5), and slightly to your left (292.5 to 337.5). A scene is then rendered as follows:

```
Straight ahead, you see a white van
Slightly to your right, you see a red brick building, a scaffold...
```

It is important to note that Touchdown consists of multiple subtasks, such as finding the correct scene, stopping at the correct scene, and then finally orienting to face a hidden object in the scene. In this work, similar to Zhong et al. [49], we only consider the first task of finding the correct scene according to the instruction. To compare our verbalization with prior work, we also evaluate a separate setting (e.g. used in Chen et al. [12], Schumann et al. [34]) where the agent must identify when to stop and is credited so long as it stops within one panorama of the target scene. Behaviour cloning using our verbalization technique results in 63.0% task-completion rate.

**Statistics of verbalized environments**  In Appendix D Table 6, we show statistics of verbalized environments as quantitative evidence of their challenges.

# E Pseudocode for Policy Improvement using Language Feedback Models

In this section we detail, using pseudocode, the procedure for policy improvement using Language Feedback Models. Algorithm 1 describes learning a model from LLMs. Algorithm 2 describes identifying desirable behaviour that are productive for solving tasks specified in the instruction, and then using this behaviour for imitation learning. Algorithm 3 describes the iterative policy improvement procedure using these two algorithms.

---

**Algorithm 1** TRAINFEEDBACKMODEL: Training a Language Feedback Model using LLM feedback.

---

1: Inputs: initial policy $\pi$, LLM LLM, environment $E$
2: Feedback dataset $F \leftarrow \{\}$
3: **for** $i = 1 \ldots N$ **do**
4:     $x \leftarrow$ SAMPLEINSTRUCTION
5:     $\tau_i \leftarrow$ ROLLOUT$(\pi, E, x)$
6:     **for** window $w_j$ in $\tau_i$ **do**
7:         $y \leftarrow$ QUERYLLMFORFEEDBACK$($LLM$, w_j, x)$
8:         **for** verbalized observation $v_k$, LLM feedback $y_k$ in each step of $y$ **do**
9:             $F \leftarrow F \bigcup (v_k, y_k)$
10:         **end for**
11:     **end for**
12: **end for**
13: Feedback model $f \leftarrow$ TRAINLM$(F)$

---

**Algorithm 2** IMITATEUSINGFEEDBACK: Imitation learning using desirable behaviour identified by a feedback model.

---

1: Inputs: base policy $\pi$, environment $E$, feedback model $f$
2: Imitation dataset $G \leftarrow$ behaviour cloning dataset
3: **for** $i = 1 \ldots N$ **do**
4:     $x \leftarrow$ SAMPLEINSTRUCTION
5:     $\tau_i \leftarrow$ ROLLOUT$(\pi, E, x)$
6:     **for** verbalized observation $v_k$, action $a_k$ in each step of $\tau_i$ **do**
7:         $y_k = f(v_k)$
8:         **if** $y_k$ is desirable **then**
9:             $G \leftarrow G \bigcup (v_k, a_k)$
10:         **end if**
11:     **end for**
12: **end for**
13: Improved policy $\pi' \leftarrow$ TRAINLM$(G)$

---

**Algorithm 3** Policy improvement using Language Feedback Models.

---

1: Inputs: base policy $\pi$, environment $E$
2: Feedback model $f \leftarrow$ TRAINFEEDBACKMODEL$(\pi,$ LLM$, E)$
3: $\pi_0 \leftarrow \pi$
4: **for** $k = 1 \ldots N$ **do**
5:     $\pi_k \leftarrow$ IMITATEUSINGFEEDBACK$(\pi_{k-1}, E, f)$
6: **end for**

---

Table 7: Task completion rate on evaluation benchmarks, including DAGGER.

|  | ALFWorld | ScienceWorld | Touchdown |
|---|---|---|---|
| BC | 62.6 | 45.8 | 57.5 |
| ACTPRED | 56.0 | 39.0 | 58.0 |
| DAGGER | 55.2 | 22.5 | 50.2 |
| LFM | **64.1** | **47.1** | **59.7** |
| LFMA | 74.6 | 49.3 | 62.8 |

## F  Comparison to DAGGER

Our main experiments in Section 5.4 illustrate the difficulty of using LLMs as an expert to predict actions. Specifically, we show that when these predictions are used for imitation learning, the resulting policy improvement is worse than using Language Feedback Models. This performance degradation is exacerbated in environments with larger action spaces, such as ScienceWorld.

DAGGER [31] is an intermediate method between Language Feedback Models and using LLMs as an expert policies for imitation learning. Specifically, in DAGGER, we also use LLMs as experts to predict action. However, instead of using LLMs during each step, in DAGGER we use LLMs to provide batched retroactive action prediction similar to how in Language Feedback Models we use LLMs to provide batched retroactive feedback. Here, we apply DAGGER action prediction to the exact same number of examples as when we collect feedback data for LFMs. In Table 7, we compare DAGGER performance to those using LLM as an expert (ACTPRED) and using Language Feedback Models (LFM). We find that although DAGGER is more efficient than ACTPRED in that it annotates synthetic examples in batch, it underperforms ACTPRED (and consequently LFM) across all benchmarks.

Table 8: Agreement between GPT-4 and LLAMA 2 across the benchmarks. We collect steps from rollouts on the training environments where either GPT-4 or LLAMA 2 identified a productive action. This table shows percentage of of those actions that are identified exclusively by GPT-4, exclusively by LLAMA 2, and identified by both models. The total number of steps identfied are 40569 for ALFWorld, 68565 for ScienceWorld, and 90529 for Touchdown.

| | GPT-4 only | LLAMA 2 only | both |
|---|---|---|---|
| ALFWorld | 14.4% | 49.3% | 36.2% |
| ScienceWorld | 10.2% | 62.3% | 27.5% |
| Touchdown | 22.3% | 67.3% | 10.4% |

Table 9: Human verification of LLM feedback in terms of percentage of true positives and false positives. A true positive (TP) is a step that is correctly identified by the LLM as being productive to solving the task. A false positive (FP) is a step that is wrongly identified by the LLM as productive. We manually evaluate 10 examples from each benchmark, each with up to 20 steps. Support (# of steps) is shown in brackets.

| | GPT-4 | | LLAMA 2 | |
|---|---|---|---|---|
| | TP | FP | TP | FP |
| ALFWorld | 100% (22) | 0 | 32% (18) | 68% (38) |
| ScienceWorld | 78% (38) | 22% (11) | 48% (38) | 52% (41) |
| Touchdown | 81% (22) | 19% (5) | 39% (24) | 61% (38) |

# G   Quantitative and Qualitative Analyses of Learned Language Feedback

**Comparison of GPT-4 to LLAMA 2 70B**   How much difference is there between language feedback obtained from the open-source LLAMA 2 vs from GPT-4? Table 8 shows that, surprisingly, there is a large degree of disagreement between GPT4 and LLAMA 2. Specifically, LLAMA 2 identifies significantly more actions as being productive to achieving the goal.

We perform a manual analysis of language feedback by GPT-4 and LLAMA 2 to characterize qualitative differences between feedback collected by these two models. First, we roll out BC policies, then ask each model for feedback. Each example contains a segment of up to 20 steps extracted from a rollout, and the LLM is prompted to list productive steps. For each step the LLM identifies as productive to solving the task, we manually verify whether the step is indeed productive. We manually inspect 10 examples from each model for each benchmark, for a total of $10 \times 2 \times 3 = 60$ examples. Table 9 shows the number of true and false positives predicted by both models in this manual evaluation. We find that a significant number of steps are incorrectly determined by LLAMA 2 as desirable. When we train the policy on a combination of LLAMA 2 data and demonstrations used to learn the BC policy, we obtain worse task-completion percentage than using GPT-4 data and demonstrations. Specially, performance drop from 64.1% (GPT-4) to 56.0% (LLAMA 2) on ALFWorld, from 47.1% to 47.0% on ScienceWorld, and from 59.7% to 56.5% on Touchdown.

Table 10 shows some examples of steps identified as productive by these models that illustrate LLAMA 2's tendency to identify spurious actions as being productive. In the ALFWorld examples, for instance, LLAMA 2 has a strong tendency to identify opening and closing cabinets and drawers as productive, even though they have nothing to do with putting a clean soap bar on the counter top (the first instruction) or putting a clean spatula on the side table (the second instruction). Similarly in ScienceWorld, LLAMA 2 identifies unnecessary actions such as going outside (example 1) and going to the bedroom (example 2) as productive, even when the instruction explicitly details that the aluminum foil is found in the kitchen (example 1) and that the unknown substance is found in the workshop (example 2). Finally, LLAMA 2 also tends to identify spurious actions as productive in Touchdown. In the last example, the instruction asks to take a left after the first intersection, but LLAMA 2 rewards the left turn during the first turn, before the agent even arrives at the first intersection. GPT-4, on the other hand, correctly identifies Step 8, when the agent finally encounters the first intersection, as productive.

We show in Section 5.4 that small and cost-effective Language Feedback Models are able to replicate LLM feedback through training. Our comparison between GPT-4 and LLAMA 2 show that a less powerful model such as LLAMA 2 are unable to provide high-quality feedback. The summary from this experiment are then that 1) powerful LLMs are necessary to provide good feedback, but expensive to run during on-line policy improvement 3) consequently, learning small LFMs is an effective solution to achieve high feedback performance while reducing inference cost during policy improvement.

## H   GPU Usage

We train feedback models and policies using 80GB A100 GPUs. To produce rollouts at in parallel, we use a cluster of 200 32GB V100 GPUs. For all environments, feedback model training takes under 24 hours using one A100 GPU while inference can be performed locally using a 32GB GPU under 2 hours. Policy training requires 1 day for ALFWorld, 2 days for ScienceWorld, and 3 days for Touchdown. For all environments, policy rollout over the entire evaluation environments can be performed over the cluster of 200 32GB V100 GPUs in under 6 hours.

# I  Correlation between LLM/LFM feedback and true state values

To investigate whether LLM feedback is correlated with true state values, we obtained partial rollouts for environments in ALFWorld and asked GPT4 to score from 1-5 whether the partial rollouts are on the right track to solving the task. We then ran a planner (with full observability) to complete these partial rollouts in order to obtain ground truth optimal values. With no training, the LLM's predicted score has strong correlation (0.61 Pearson) with the optimal values. This means that GPT as a feedback model has strong zero-shot generalization on environments it was not trained on.

In addition, we regressed a FLAN-T5 model to estimate state values from language feedback using states from a random policy, then evaluated its predictions against true state values on mixed policies where an expert is select $p$ fraction of the time (and a random policy is used other times). When we uniformly sample states from the random policy, for $p = 0, 0.2, 0.4, 0.6, 0.8$, the regression model achieves Pearson correlations of $0.3 - 0.4$. When we subsample training states evenly across values, we achieve Pearson $0.65 - 0.75$. This shows that state coverage, for which state value coverage is a proxy, is indeed important. Conversely, when we use mixed policies for training as well, we obtain $0.78 - 0.85$ Pearson. This shows that optimality is indeed important.

Table 10: Example steps identified as productive by GPT-4, LLAMA 2, and both. Touchdown steps are truncated for brevity.

**ALFWorld**

Task: clean some soapbar and put it in countertop.
Step 3 Your action: take soapbottle 2 from toilet 1 Result: You pick up the soapbottle 2 from the toilet 1.
Step 6 Your action: go to countertop 1 Result: You arrive at loc 11. On the countertop 1, you see a candle 2...
Step 11 Your action: open cabinet 1 Result: You open the cabinet 1. The cabinet 1 is open. In it, you...
Step 17 Your action: open cabinet 1 Result: You open the cabinet 1. The cabinet 1 is open. In it, you see...
Step 19 Your action: go to cabinet 3 Result: You arrive at loc 12. The cabinet 3 is closed.
Task: put a clean spatula in sidetable.
Step 3 Your action: go to sidetable 1 Result: You arrive at loc 22. On the sidetable 1, you see nothing.
Step 10 Your action: go to drawer 3 Result: You arrive at loc 19. The drawer 3 is closed.
Step 11 Your action: open drawer 3 Result: You open the drawer 3. The drawer 3 is open. In it, you...
Step 14 Your action: open drawer 4 Result: You open the drawer 4. The drawer 4 is open. In it, you...
...

**ScienceWorld**

Task: Your task is to determine if aluminum foil is electrically conductive. The aluminum foil is located around the kitchen. First, focus on the aluminum foil. If it is electrically conductive, place it in the yellow box. If it is electrically nonconductive, place it in the purple box. Do not focus on objects not central to completing the task!
Step 21 Your action: open door to greenhouse Result: The door is now open...
Step 23 Your action: open door to outside Result: The door is now open...
Step 25 Your action: open door to foundry Result: The door is now open...
Step 29 Your action: go to outside Result: You move to the outside...
Step 31 Your action: go to kitchen Result: You move to the kitchen...
Step 32 Your action: pick up aluminum foil Result: You move the aluminum foil to the inventory...
Step 33 Your action: focus on aluminum foil Result: You focus on the aluminum foil...
Step 37 Your action: go to workshop Result: You move to the workshop...
Step 38 Your action: move aluminum foil to yellow box Result: You move the aluminum foil to the yellow...
Task: Your task is to determine if unknown substance N is electrically conductive. The unknown substance N is located around the workshop. First, focus on the unknown substance N. If it is electrically conductive, place it in the blue box. If it is electrically nonconductive, place it in the orange box. Do not focus on objects not central to completing the task!
Step 1 Your action: open door to art studio Result: The door is now open...
Step 4 Your action: go to hallway Result: You move to the hallway...
Step 6 Your action: go to bedroom Result: You move to the bedroom...
Step 9 Your action: open door to hallway Result: The door is already open...
Step 11 Your action: open door to workshop Result: The door is now open...
Step 12 Your action: go to workshop Result: You move to the workshop...
Step 13 Your action: pick up unknown substance Result: You move the unknown substance N to the...
Step 15 Your action: move unknown substance to blue box Result: You move the unknown substance N to...

**Touchdown**

Task: Going with the flow of traffic, proceed to the intersection and take left. There should be some blue scaffolding on your right hand side, take a left here. You should now have a tall solid green fence on you right hand side, go straight to the next intersection and take a left. There's a bunch of scaffolding on the left hand side, go straight. Pass the first intersection you come across. You will pass mail trucks parked on both sides of the road.
Step 1 Your action: slightly to your left Result: behind you, you see: parked, white box trucks , two wider sidewalks , this narrow two lane road , the right sidewalk buildings . behind you, sightly to your left, you see: , three air-conditioners , three awning , a smaller yellow taxi . to your left, you see: , the theater awning , a yellow cab car , the second purple awning . slightly to your left, you see: , a white-capped hydrant , ornate gray balconies , the purple wayfair truck . straight ahead, you see: , a median strip , some tall, brick buildings , parked, white box trucks , a blue bus lane sign , the right sidewalk buildings . ...
Step 8 Your action: straight ahead Result: behind you, you see: , a brown storefront , surface streets , the right sidewalk buildings , a parked black box truck , some unremarkable brick buildings . behind you, sightly to your left, you see: , then a storefront , a white/grey van , a large, blocky, gray building . to your left, you see: , a large white store sign , a construction vehicle , a long gray and white 5 story building slightly to your left, you see: , some tall, brick buildings , fedex van , a white/grey van . straight ahead, you see: , a large red brick apartment building , an orange and white traffic object , 3rd and 4th intersections , a small blue car , the right sidewalk buildings , the parked yellow suv taxi . slightly to your right, you see: ...

