# OpenReview forum: "Policy Improvement using Language Feedback Models"
_NeurIPS.cc/2024/Conference — NeurIPS 2024 poster_

### Official Review · Reviewer_B45L · 2024-07-05

**Soundness:** 3
**Presentation:** 3
**Contribution:** 2
**Rating:** 7
**Confidence:** 3

**Summary:**

The authors introduce Language Feedback Models (LFM), a method filtering out desirable transitions (i.e. transitions considered as helping to solve a task) collected by an agent in an environment to then improve this agent by doing Imitation Learning on these transitions. The LFM method has three phases: 1) First, a fairly large number of transitions are collected by the initial policy, and GPT4 is used to determine which transitions are desirable. Then, 2) given this dataset of desirable transitions, a smaller LLM is trained to reproduce GPT4 in saying these transitions are desirable. Finally, 3) the policy is used to collect new transitions, which are then filtered out by the smaller LLM, taking only the ones considered as desirable, and the policy maximizes the likelihood of the actions chosen in these transitions.

Experiments in two TextWorlds and one visual navigation environment are performed, showing that LFM successfully improves the initial policies (learned with Behavioral Cloning from expert demonstrations) and leads to better results than using GPT4 to directly propose the best action to perform and imitate it. Additionally, experiments show that the learned feedback model can be used to improve the policy in unseen test tasks which further increases the policy's performance even though the feedback model has not been trained on the test tasks. The authors also show that one can go further and not only train the smaller LLM to select desirable transitions but also to explain why these transitions are desirable (also by imitating GPT4), leading to better explainability.

**Strengths:**

The method relies on distilling the feedback ability from GPT4 to a much smaller LLM (Flan-T5 770M) to reduce the cost of asking for feedback at every step. The authors show that they successfully transferred this ability and that the obtained feedback model generalizes to unseen tasks. The experiments also show that their method is not only less compute-effective but also much more efficient than directly imitating GPT4 used as the policy.

The authors also show the method can be trained to provide explanations of the feedback given, leading to better explainability of the method.

**Weaknesses:**

One of the main weaknesses I see is the lack of baselines to properly assess the efficiency of the method. The LFM method acts like a rejection sampling algorithm where only transitions considered desirable by the feedback model are kept to fine-tune the policy. Similarly, If all environments provide a reward if would have been interesting to have a baseline relying on this information for the rejection sampling. If the reward is sparse, one could wait for the end of the episode and only keep desirable trajectories (e.g. whose episodic reward is above some threshold or if the goal has been reached).

The authors also say "In this work, we consider long-horizon settings with only sparse and delayed task-completion rewards. Consequently, we focus on imitation learning from demonstrations as opposed to reinforcement learning from rewards." It could also be interesting to see how RL performs with the same amount of collected data.

Finally, it seems the authors use a dataset of expert demonstrations to first train the initial policy with BC before applying their method and the compared baselines. However, they do not provide any insights on how important this preliminary phase is. Indeed, on key advantage of LFM is that it relies on BC while not requiring an expert policy. However, results from Table 3 show that LFM mostly seems to substantially improve the initial policy learned with BC from expert demonstrations.

**Questions:**

- In Table 3, BC results are different from "Prev SOTA" on ALFWorld and Touchdown. Why is this the case, and how did the authors collect expert demonstrations that led to a policy better than SOTA?
- When describing the ACTPRED baseline l237, it is mentioned "First, we execute k steps of the base policy, then query the LLM for the next action". What are these k first steps and why are they necessary?
- The authors computed the perplexity averaged over tokens of each possible action with the policy (that is an LLM) and selected the action with the minimum perplexity. Why did the authors not use the (log) probability of the sequence of tokens as in SayCan (Ahn et. al, 2022), GLAM (Carta et. al 2023) or TWOSOME (Tan et. al 2024)?
- Also, computing the perplexity (or probability) of each action to follow the prompt can be very extensive when the action space grows, given that the policy is still a fairly large model (770M). Did the authors use anything to make this fast enough, especially in ScienceWorld where the set of possible actions can be very large?
- In LFM, an initial phase uses the policy to collect trajectories that are then given to GPT4 in order to obtain the dataset to finetune the feedback model. In the experiments, 10k trajectories of 20 steps have been collected if I understood well. Were these trajectories also appended to the dataset used to improve the policy later on?
- The authors showed an example of an explanation provided by the feedback model (based on Flan-T5 770M) in Table 5. Given the limited abilities in text generation of Flan-T5 770M, did the authors perform a deeper study of the explanations provided by the model, especially on the unseen test tasks?

**Limitations:**

The authors discuss the limitations of the introduced method in the appendices. They identify that, while LFM improves compute efficiency by distilling feedback abilities from GPT4 into a smaller LLM, Flan-T5 770M is still a fairly large model to be called at every step, both for the policy and the feedback.

Also, Appendix G shows that using Llama 2 70B instead of GPT4 can significantly impact the results. This is not a strong limitation, in my opinion, given that GPT4 is only used in the initial phase to train the smaller LLM.

---

> ### Author Rebuttal · Authors · 2024-08-03
>
> We thank the reviewer for taking the time to review our paper and provide us with valuable insights. We appreciate the acknowledgement of the generalizability and efficiency of our method, as well as improvements to model explainability.
>
> ## W1: Rejection sampling
> We thank the reviewer for drawing connections between our method and rejection sampling (RS). We discuss 2 key differences between the two techniques.
>
> First, LFM removes 50-80% of non-productive steps whereas RS includes entire trajectories (20-100 steps for our envs). That said, we experimented with training on entire rollouts w/ positive reward. This did not significantly improve over the base policy.
>
> Second, LFM filtering can be done during testing, where no reward is present. In this case, reward-based RS cannot be directly applied to perform adaptation during testing. That said, one could train a model to predict rejection, using data from the training environments, then use said model on test environments for adaptation. This is precisely what our proposed method does.
>
> ## W2: reinforcement learning
> We reference prior RL work on ALFWorld and Touchdown (https://arxiv.org/abs/2110.10661), which shows RL underperforms LFM after training for 10 million steps. ALFWorld episodes typically have <30 steps. Training on demos amounts to 30 * 3.5k ~ 100k steps. We train LFM using 1 rollout per training env for another 100k steps. Touchdown episodes are typically <200 steps. Demos and LFM steps are then 1.5 million steps each. For both envs, LFM requires fewer steps than RL and achieve higher task success rate (e.g. 64 vs 23 ALFWorld, 60 vs 15 Touchdown).
>
> ## W3: Importance of initial demonstrations
> We experimented w/ a random base policy, but found collecting trajectories that demonstrate productive behaviour difficult, as random policies are mostly not productive. Consequently, most data given to the LLM for feedback annotation do not exhibit productive behaviour, therefore increasing the cost of LLM usage. We can alleviate this using reward-based RS, as the reviewer suggested in W1. That is, we sample many trajectories, then give those w/ non-trivial reward to the LLM for feedback annotation. We will explore this in future work.
>
> ## Q1: BC vs Prev SOTA
> We use the same demos as prior SOTA, provided by the env designers. However, we use stronger, newer base models. For instance, ALFWorld prior SOTA fine-tuned a GPT2 model. Touchdown prior SOTA trained a CNN/RNN network. In contrast, we fine-tuned a FLAN T5 Large model, which performs well in language grounding tasks (https://arxiv.org/abs/2210.11416). We perform similarly to prior SOTA for ScienceWorld using the same model but without ScienceWorld-specific prompts (e.g. tracking object types, rooms visited).
>
> ## Q2: ActPred k steps
> The significance of this k-step is due to the token limitation. We want diverse trajectories over many envs. However, each trajectory may consist of >100 steps (ScienceWorld and Touchdown are often 50-200 steps). Predicting actions from the start using LLMs is biased towards early-trajectory steps - we will exceed the token limitation before encountering late-trajectory steps. Completely rolling out one env at a time will exceed token limitation before processing many of the envs. To balance between coverage of envs and trajectory depth, we first roll out k steps, then predict action using LLM. k is sampled from the max trajectory length of demos.
>
> Note that this coverage problem is less significant for LFMs. Because LFM data is queried in windows of 20-steps, we can maintain high coverage of both environments and of trajectory depth. We also experimented with asking GPT4 to retroactively relabel actions in 20-step windows, however the labels were significantly worse than labeling actions one step at a time.
>
> We will add explanations of the k-step significance to the manuscript.
>
> ## Q3: average perplexity vs logprob of sequence
> We did explore with log probability, which performed slightly worse than averaged perplexity.
>
> ## Q4: Scoring large action spaces
> We emphasize that training strictly maximizes log probability of tokens of the correct action, so its complexity is independent of the size of the action space. Inference requires scoring large action sets, however is cheap due to lack of gradient-tracking. The only optimization we did was compute scores in chunks, which are concatenated to select the optimal action.
>
> ## Q5: initial phase trajectories
> We clarify that it is 10k windows of 20 steps sampled from trajectories. These windows contain many negative examples, consequently we do not add them to the dataset for policy improvement. The data used for policy improvement are only steps from base policy rollouts that identified by the trained LFM as productive.
>
> ## Q6: analysis of explanations
> The examples in Table 5 are actually from the held-out test set tasks. Unfortunately, we did not perform quantitative analyses. Our qualitative findings are that a 770M model, trained on LLM feedback for a specific domain, is able to provide accurate feedback and explanations for said domain. One limitation was that LFMs are sometimes not consistent in evaluating productiveness of ambiguous actions such as exploration. For instance, if the instruction is to find cups and wash them, then even a good policy will spend time searching the room for cups. To some observers, this search procedure is productive, to others, it is not productive. In our experience, LLMs and LFMs are sometimes not consistent in assigning productiveness to such actions. In future work, we would like to analyze the limits of what types of feedback LLMs (and VLMs, for that matter) are able to provide in multi-modal grounded environments.
>
> ## Summary
> We sincerely thank the reviewer for taking their time to help us improve this work. We hope we have addressed the reviewer’s concerns and questions. If so, would the reviewer please consider increasing their score to show support for our work?

---

> > ### Comment · Reviewer_B45L · 2024-08-09
> > **Response to rebuttal**
> >
> > The authors properly answered all my questions and concerns.
> > I will keep my score as it is.

---

### Official Review · Reviewer_hZ53 · 2024-07-12

**Soundness:** 2
**Presentation:** 2
**Contribution:** 2
**Rating:** 4
**Confidence:** 4

**Summary:**

This paper provides a policy improvement method with a Language Feedback Model (LFM) for decision making tasks.

The proposed method mainly consists of two stages: (1) training a Language Feedback Model (LFM) and (2) improving a policy model with the trained LFM. In the first stage, to train a LFM, the initial policy is used to generate rollouts by interacting with the environment. Then, a LLM is used to generate text feedback on each action in the rollouts. Then, a LFM is trained on this feedback dataset. In the second stage, the trained LFM is used to generate feedback on each action generated by the initial policy. Then, actions that are predicted as desirable from the LFM are collected as a dataset. Finally, the policy is trained on the dataset that contains desirable actions.

This paper evaluates the proposed method on three decision making benchmarks: Touchdown, ScienceWorld, and ALFWorld. This paper empirically demonstrates that the proposed method provides better scores than baselines such as BC or ActPred.

**Strengths:**

S1. The idea of learning a language feedback model (LFM) to provide text feedback on each action sampled from the initial policy is interesting.

S2. This paper provides experimental results on three representative benchmarks: Touchdown, SecienceWorld, and ALFWorld.

**Weaknesses:**

W1. In the introduction section, this paper mentions that sample-efficiency and generalizability are important in instruction-following agents. I am not sure that the policy improvement with LFM is sample-efficient and generalizable. The proposed method trains the initial policy on a rollout dataset where desirable actions are collected by the LFM. Training on many rollouts does not seem sample-efficient. Also, the policy trained on a specific environment may lose generalizability.

W2. In Section 3.2 (i.e., Learning a feedback model), it is rather unclear how to collect desirable behavior D_k.

W3. It would be better to provide ablation study that shows effectiveness of the proposed method. For example, the authors may compare the policy improvement with an expert LLM to the policy improvement with the LFM.

**Questions:**

Q1. Regarding the weakness W1, how many rollouts are needed to train a LFM? Also, how many revised rollouts are need to train an initial policy?

Q2. Regarding the weakness W2, if we only select desirable actions from a rollout, does the revised trajectory correctly reflect or follow the environment dynamics or the transition function?

**Limitations:**

The authors provide limitations of their work in Appendix A (i.e., Limitations).

---

> ### Author Rebuttal · Authors · 2024-08-03
>
> We thank the reviewer for taking the time to review our paper and provide us with valuable insights. We appreciate the acknowledgement of the novelty of our method, as well as clear demonstrations of improvements using our method across three representative benchmarks.
>
> ## W1: sample-efficiency and generalizability
> We define sample-efficiency to mean that the agent requires few human labeled demonstrations in order to achieve high performance. In this work, we assume access to a small set of human labeled demonstrations to train a base policy. With no further human labeled demonstrations, we can synthesize demonstrations using the LFM automatically, with which we can improve policy performance. Consequently, we describe the proposed method as sample-efficient because it improves policy performance with no additional human labeled demonstrations.
>
> We define generalizability to mean that the agent generalizes to new environments not seen during training. Let’s consider the game NetHack. In NetHack, there is one implicit goal for the agent, which is to obtain the highest score it can, given the rules of NetHack. In other words, every NetHack agent is given the implicit instruction “get the highest score you can in NetHack, subject to the rules of the game”. In contrast, all three environments we investigate have different instructions between training and evaluation. These environments require generalization to new scenes (like NetHack) as well as to new instructions (unlike NetHack). For ALFWorld, the agent may be trained to “find and wash glasses” and “put apples in the fridge”, but during test evaluation it may be asked to “find apples, wash them, then put them on the dining table” in new rooms. Similarly, in ScienceWorld the agent is required to follow new compositional instructions it has not seen before during training, in new spaces (e.g. the determine the boiling temperature of a new substance). In Touchdown, the agent is required to navigate between new starting and end points in new neighborhoods. Consequently, we describe the proposed method as one that generalizes because it improves performance on new environments with new scenes and new instructions not seen during training.
>
> We will clarify these two points by precisely defining sample-efficiency and generalization in the manuscript. We thank the reviewer for identifying this point.
>
> ## W2: desirable behaviour
> We want to emphasize that the proposed framework consists of two components. The first component involves learning a language feedback model (LFM). The second component involves using this LFM to improve policy. The D_k the reviewer refers to has to do with the second (improving policy), not the first (learning a feedback model). We will summarize both components here.
>
> We start from a base policy trained from a small set of human demonstrations. In the first component of learning a LFM, we roll out the base policy to collect trajectories. These trajectories are given to an LLM to create a dataset (this is not D_k) to train the LFM. This dataset consists of tuples of (agent behaviour, LLM feedback). We train a smaller LFM to imitate LLM feedback. That is, given the agent behaviour, the LFM critiques in natural language whether this behaviour is productive. We then freeze this LFM.
>
> In the second component, we use the trained LFM to improve the base policy. First, we roll out the base policy to collect trajectories. We then use the LFM to predict whether each step in the trajectory is productive. We take the subset of productive behaviour (eg. context and agent action) and add it to our collection of productive demonstrations. Then we update the policy by training on this collection of demonstrations. We do this in multiple rounds, in each round K, we roll out the policy, identify good behaviour, and add it to the collection of demonstrations. We refer to the collection of demonstrations during round k as D_k.
>
> ## W3: compare expert LLM to policy improvement with LFM
> We do perform two such experiments in our ablation study. Experiment A compares to using GPT4 (0315) on-policy zeroshot. Table 3 shows that our method (and other methods that train a smaller policy model) achieves significant improvements over zero-shot GPT4. Experiment B uses GPT4 to label what the agent should do, as opposed to critique how the agent performed. We then train a policy using demonstrations as well as GPT-4 labeled actions. This method of using LLM as expert policy to label actions (ActPred), underperforms using LLMs to train feedback models (LFM).
>
> ## Q1: number of rollouts
> Dataset statistics are shown in Appendix C Table 6. To train the initial policy, we use 3.5k demos for ALFWorld, 3.6k for ScienceWorld, and 6.5k for Touchdown. These correspond to 1 demo for each unique environment. When rolling out, we perform one rollout per unique environment. Consequently we have an identical number of rollouts as initial demos. That is, we perform 3.5k rollouts for ALFWorld, 3.6k for ScienceWorld, and 6.5k for Touchdown.
>
> ## Q2: desirable actions vs. transition function
> The revised trajectories correctly reflect env dynamics because they are steps that took place during the rollout, according to env dynamics. This observation also explains underperformance of ActPred. In ActPred, an LLM is used to label what actions the agent should take. However, the LLM can hallucinate actions, resulting in demonstrations that do not correctly reflect env dynamics. In contrast, our proposed method only identifies real actions that took place as productive behaviour. Consequently all demos collected using LFMs correctly reflect environment dynamics. This also explains why LFM improvement is larger than ActPred.
>
> ## Summary
> We sincerely thank the reviewer for taking their time to help us improve this work. We hope we have addressed the reviewer’s concerns and questions. If so, would the reviewer please consider increasing their score to show support for our work?

---

> ### Author Response · Authors · 2024-08-12
>
> Dear Reviewer,
>
> We wanted to send a friendly reminder that we are awaiting your response. With the deadline for the reviewer-author discussion approaching on August 13, we would greatly appreciate it if you could provide feedback at your earliest convenience. We hope we have addressed your concerns and questions. If so, would you please consider increasing your score to show support for our work?
>
> Sincerely,
> Authors

---

> ### Comment · Reviewer_hZ53 · 2024-08-13
> **After the Author Response**
>
> Thank you for providing thoughtful responses to my comments. For now, I maintain my initial rating. However, I am open to AC's decision on this paper.

---

> > ### Author Response · Authors · 2024-08-13
> >
> > Thank you for your acknowledgement! Is there anything unsatisfactory about our response, such that the you would not consider increasing your score? Specifically, 1) we provided clarifications regarding sample-efficiency and generalizability, 2) we elaborated on how desirable behaviour is collected, and 3) we noted that the existing manuscript does contain comparing to expert LLMs as the reviewer requested. Is there anything else the you would like to discuss in order for you to support our work?

---

### Official Review · Reviewer_u2Y5 · 2024-07-12

**Soundness:** 3
**Presentation:** 2
**Contribution:** 3
**Rating:** 6
**Confidence:** 4

**Summary:**

The papers present a method to essentially filter which actions should be used to learn a policy via imitation learning. The method follows the online imitation learning setting and replaces the expert policy with a language feedback model (LFM) distilled from a LLM. The LFM evaluates which transitions from the policy's rollouts should be used to train the policy. A LFM is used instead of LLM to reduce the computational complexity of the task, and both consumes and produces text. The method is evaluated on several benchmarks. The method is compared against several ablations over the design decisions.

**Strengths:**

- The paper proposes an interesting idea to filter out the data that should be used for imitation learning, and then proposes to do this with a language model.
- The method demonstrates clear improvements to methods evaluated against.

**Weaknesses:**

- The authors claim that their LFM method is better than using a LLM as an expert policy, because it scan provide human interpretable feedback. However, the authors do not provide any results to suggest that the LFM can produce outputs humans can do something with.
- The only comparisons are to ablations of the proposes method. At a minimum, some of the mentioned related work (like MOTIF (Klissarov et al [22])) should probably be a baseline.
- The LMFA 1 rnds and LMFA 2 rnds in Table 3 do not seem to be discussed in the main body.
- There are some gaps writing that have left me with a lot of questions/uncertainties (see below).

- Small things:
     - you have places in the PDF with weird formatting, e.g. lines 156 - 157.
     - The reference to Figure 1(b) in Section 2 line 72 should probably be 1(c)

**Questions:**

- It is unclear how the method is marking states as desirable in section "learning from language feedback" is different from that in "naively learning from LLM feedback". Is there a connection missing that links how the LFM is "efficiently" learned and how the policy learns? Instead of querying the LLM at each step, the LLM is queried of a trajectory of steps and responds by saying which steps were desirable?
- Is there also a reward function?
- What is an iteration/round in this method? i.e. "In round k, we rollout the base policy...." (Section "Learning from language feedback")
- You state, "Unlike these works ... we consider settings where training and evaluation goals are different." It is not clear to me how the training and evaluation goals differ in your set up.
- How do you decide when to summarize what has happened with a "Before" in the LLM prompt? e.g. Table 2 - LFM prompt.
- What does it mean that you "...limit the amount of LLM usage to 100k GPT-2 tokens"? Did you constrain the number of input tokens? The number of output tokens? How did you go about the constraints?
- How did you select 20 as the number of timesteps over which LLM feedback was collected?
- You say you subselect feedback data to have an even split of productive and non-productive action. How much data do you actually end up with? What were the originally ratios? How diverse are the samples?
- For the results section on generalization to new environments, what makes a new environment new? Table 4 holds results for ALFWord, ScienceWorld, and Touchdown, which are the environments you report training on.

**Limitations:**

There are no limitations in the main body.

---

> ### Author Rebuttal · Authors · 2024-08-03
>
> We thank the reviewer for taking the time to review our paper and provide us with valuable insights. We appreciate the acknowledgement of the novelty of our method, as well as clear demonstrations of improvements using our method.
>
> ## W1: interpretable feedback
> To clarify, we have 2 claims. First, policy improvement from LLMs feedback is better than using LLM as policy. There are two experiments that support this. a) we compare to using GPT4 (0315) on-policy zeroshot. Table 3 shows that our method achieves significant improvement. b) we use GPT4 to label what the agent should do, then train a policy using demos as well as GPT-4 labeled actions. This method of using LLM as expert policy to label actions (ActPred), underperforms using LLMs to train feedback models (LFM).
>
> The interpretable feedback experiments show that instead of producing LFMs that only identify good behaviour, we can train descriptive LFM-Ds that also state why a behaviour is good. We show that LFM-Ds perform similarly to LFMs, achieving both interpretability and high levels of policy improvement.
>
> ## W2: related work
> Thank you, we will add this. LFM differs from MOTIF in that the former improves policies via imitation learning while the latter derives a reward model for RL. While MOTIF shows results on a single game NetHack, we show results across three different settings in household (ALFWorld), scientific experiments (ScienceWorld), and real-scene navigation (Touchdown). In the latter two settings, RL is  substantially worse than LFMs (https://arxiv.org/abs/2110.10661). In the future, we will investigate RL using language feedback for multimodal grounded settings.
>
> ## W3: adaptation rounds
> Thank you for the comment. We describe one-round adaptation in section 5.2 on line 249. We will describe two-round adaptation in the manuscript. Round-wise adaptation is described on ​​line 148: we use the policy from the previous round to perform rollouts, then filter for good behaviour using the trained LFM, then imitate them to improve the policy.
>
> ## W4: small things
> Thank you, we will make the corresponding corrections to the manuscript.
>
> ## Limitations
> We discuss limitations and broader impacts of this method in Appendix Section A and B, we will move them before references.
>
> ## Q1: efficient vs. naive feedback learning
> Let’s suppose we have 20 steps. The "naive" method queries the LLM each step, resulting in 20 queries. The kth query has steps 1…k-1 in the context and asks the LLM whether the kth action is productive. The "efficient" method batches feedback requests into 1 query, which asks the LLM to list which steps were productive (Figure 2). The efficient method is much cheaper, requiring 20x fewer API calls to the LLM (line 130). Once feedback is collected, the subsequent training would be identical.
>
> ## Q2: reward function
> We do not train a reward function - we only use imitation learning for policy improvement, not reinforcement learning.
>
> ## Q3: iteration/rounds
> After training a LFM, we can improve the policy in rounds. In round 1, we start with the base policy P1 trained on initial demos. We roll out P1, and then identify its good behaviour using our LFM. We then add the good behaviours into the demo set and train the policy P2. We then repeat this in round 2, where we identify good behaviours using P2, then use those to train P3, and so on.
>
> ## Q4: training vs evaluation goals
> Let’s consider NetHack. NetHack has 1 implicit goal for the agent, which is to obtain the highest score it can. In contrast, we consider settings w/ different instructions between train and test, which require generalization to new scenes (like NetHack) and new instructions (unlike NetHack). For ALFWorld, the agent may be trained to “find and wash glasses” and “put apples in the fridge”, but during test it may be asked to “find apples, wash them, then put them on the dining table” in new rooms. Similarly, in ScienceWorld the agent is required to follow new instructions in new spaces (e.g. the determine the boiling temperature of a new substance). In Touchdown, the agent is required to navigate between new starting and end points in new neighborhoods.
>
> ## Q5: summarize with “Before”
> In Table 2, the “Before” is not a summary, it is the observation of the step right before the window starts. In this case step 20. We will clarify this in the manuscript.
>
> ## Q6: Token limitation for LLM usage
> As described in Table 3, we limit LLM interactions to 100k output tokens. We do not limit input tokens as they are much cheaper than output tokens. We collect feedback for as many windows as possible until we run out of 100k output tokens, then we use this feedback to train the LFM. For ActPred, we label actions for as many steps as possible, until we run out of 100k output tokens, then we use this annotated set along with demos to train the policy. We will clarify this in the manuscript. Thank you!
>
> ## Q7: 20 steps
> Empirically, 20 steps fits over 90% of observation windows into the model’s context length (8k for GPT4 0315). We will investigate using new LLMs w/ longer context (128k) to train LFMs.
>
> ## Q8: Data ratios and sample diversity
> This ratio differs between settings (60% not productive for ALFWorld, 70% ScienceWorld, 90% Touchdown). All settings use 10k 20-step windows as feedback data.
>
> We have diversity across tasks, instructions, and envs. LFM data collection is biased by how good the base policy is. If the base policy is bad at task A and good at B, then we tend to identify more “good behaviour” from trajectories from B. In this work, we do not do sophisticated filtering to rebalance on a task level, but we are interested in exploring this in the future.
>
> ## Q9: new environments
> Please see Q4.
>
> ## Summary
> We sincerely thank the reviewer for taking their time to help us improve this work. We hope we have addressed the reviewer’s concerns and questions. If so, would the reviewer please consider increasing their score to show support for our work?

---

> > ### Comment · Reviewer_u2Y5 · 2024-08-09
> >
> > Thank you for your responses. You have addressed my questions, and I will raise my score.

---

### Official Review · Reviewer_8DkG · 2024-07-22

**Soundness:** 3
**Presentation:** 3
**Contribution:** 2
**Rating:** 5
**Confidence:** 4

**Summary:**

The paper proposes to train a language feedback model and leverage the language feedback model to conduct policy improvement in language-based task. The authors also propose a pipeline that can apply CLIP to convert images into textual description. Experiments over ALFWorld, ScienceWorld and Touchdown validate the effectiveness of proposed algorithms.

**Strengths:**

1, The paper is overall clear and easy to follow.
2. The idea of training a separate language model to serve as language critic function for policy learning is overall novel in decision-making task.
3. The experiments cover a large range of tasks, including one visual task -- which is great to demonstrate the generality of the proposed algorithms.

**Weaknesses:**

1. The improvement is mainly from distilling language feedback from stronger models (such as GPT-4), which somehow limits the technical contribution of proposed algorithm. Is it possible to derive language feedback from exactly the same model? (FLAN-770M might be impossible, but what about larger one like llama-3-8b?)
2. The experiments are not comprehensive enough. For instance, the table 3 presents ALF-world's SOTA from results in paper from 2021 -- which I believe is quite out-dated and there are plenty of work that improve ALF's performance a lot. Also, the authors miss several important baselines, 1. RL -- since the policy evaluation + policy improvement is the basic foundation for RL. It is essential to compare the proposed algorithms with RL. 2. a bunch of work starting after reflection, that also uses verbal feedback to improve the performance.

**Questions:**

1. There are a few literatures that might be highly relevant to the paper, including:
[1] Shinn, Noah, et al. "Reflexion: Language agents with verbal reinforcement learning." Advances in Neural Information Processing Systems 36 (2024), which also studies how to use verbal descriptions as feedback to guide the LLM's policy.
[2] Feng, Xidong, et al. "Natural Language Reinforcement Learning." arXiv preprint arXiv:2402.07157 (2024), which is also motivated by verbalizing policy learning process.
[3] A bunch of work covering llm-as-judge, like: Wang, Yidong, et al. "Pandalm: An automatic evaluation benchmark for llm instruction tuning optimization." arXiv preprint arXiv:2306.05087 (2023).

2. Why choose GPT-4-0315 to conduct experiments? This is a relatively old GPT4 model considering it's mid 2024 now? And is there any other ablation studies covering different types of llm?
3. By checking the prompt template shown in table 2, I have a question about: why there is no COT process between the model judge yes or no? Is it done on purpose or is there any explanation for that since in most setting COT can enhance GPT-4's performance -- and I believe this yes/no is very important since it directly influence the performance of policy improvement.
4. The comparison between LFMD and LFD or other baselines seems a bit unfair? Since you still require to run rollouts on test set while the other results are zero-shot.

**Limitations:**

See Weakness and Questions.

---

> ### Author Rebuttal · Authors · 2024-08-03
>
> We thank the reviewer for taking the time to review our paper and provide us with valuable insights. We appreciate the acknowledgement of the strengths of our paper, including its clarity, novelty, and demonstration of generality across a range of tasks.
>
> ## W1: improvement from distilling language feedback from stronger models
> Our primary contribution is a novel framework that combines language feedback with policy improvement. While it is true that using a single model for both policy learning and language criticism is possible, our results show that a weaker feedback model results in insignificant policy improvement (we show results using Llama2 70B in the Appendix, which is better than Llama3 8b on most benchmarks: https://github.com/meta-llama/llama3/blob/main/MODEL_CARD.md#benchmark).
>
> What our findings suggest are that
>
> 1. a very large model (let’s say an LLM) is difficult to use as a policy because it is expensive to train and slow to inference.
> 2. a small model can be trained to provide a reasonable tractable policy, however it is not capable of providing high quality feedback (without training on such feedback).
>
> Our framework uses existing large LLMs to provide high quality feedback, without further training, to improve small tractable policies for specific environments we care about.
>
> ## W2: ALFWorld baselines
> We also appreciate the feedback on the comprehensiveness of our experiments. Although Table 3 presents a paper from 2021, it is the best-performing imitation learning method without external knowledge on ALFWorld according to the ALFWorld authors. As mentioned in W1, we consider the setting where the LLM is not available during test time. Consequently we did not use reflective techniques nor CoT techniques that tend to show strong benefits with very large models (50B+, Li et al https://arxiv.org/abs/2306.14050). Are there specific works the reviewer would like comparisons to?
>
> ## W2: reinforcement learning
> We reference prior results that use RL on ALFWorld and Touchdown (https://arxiv.org/abs/2110.10661). Crucially, we show that in these challenging environments, RL underperforms LFM (and the baseline model) after training for 10 million steps. In contrast, ALFWorld episodes typically have <30 steps. Training on the demonstration dataset amounts to 30 * 3.5k ~ 100k steps. LFM improvement using one rollout per training environment, as is the case in our experiments, results in another 100k steps. For Touchdown, episodes are typically <200 steps. Demonstration steps and LFM improvement steps are consequently 1.5 million steps each. For both these cases, imitation learning and LFM improvement require substantially fewer steps than RL and achieve substantially higher task success rate (e.g. 64 vs 23 ALFWorld, 60 vs 15 Touchdown).
>
> ## W2: reflection
> We are very interested in adapting LFM to RL and with reflection, as the reviewer suggested. In this preliminary work, we wanted to scope our problem to investigate the most fundamental setting where the large model (the LLM LFM) is not used during policy improvement. Improving policies with LLMs in the loop (e.g. to provide rewards, to provide reflection) results in hundreds of steps per rollout, and is very expensive for environments we consider. However, we would like to investigate this direction in future work, for instance by learning a small language reflection model in parallel with the policy.
>
> ## Q1: references
> We appreciate the reviewer’s suggestions of several relevant works, including [1], [2], and [3]. We agree that these papers share similar motivations and approaches with ours, and we will include them in our related work section. In the current manuscript, we specifically discuss the ReAct, Reflexion, and InnerMonologue line of work in the last paragraph of our related works section.
>
> ## Q2: GPT-4-0315
> Unfortunately due to internal infrastructure policy it was the only GPT-4 model widely available to us at the time of our experimentation. We also experiment with Llama2 70B, the results for which are in our appendix. In ongoing and future work, we are investigating more recent LLMs, including VLMs, as policy critics.
>
> ## Q3: Chain of Thought
> In this work, we make the assumption that the LLM is not available during test time, only training time. Current evidence suggests that CoT is only significantly helpful when the base model is sufficiently large (50B+, Li et al https://arxiv.org/abs/2306.14050). As mentioned in response to W1, a very large model (let’s say an LLM) is difficult to use as a policy because it is expensive to train and slow to inference. Consequently, in this preliminary work, we do not investigate CoT. In future work, we would like to study how to distill large base models into small models such that they are capable of providing high quality CoT. Our result on LFM-D is a first step towards this direction, where we ask the model to provide evidence for its critique, akin to asking a question answering model to provide reasoning steps for its answer.
>
> ## Q4: LFMD and LFD and baseline comparison
> We want to make an important clarification that LFM and LFM-D do not perform rollouts on the test set - they perform rollouts only on the training set for policy improvement. The only method that performs improvement via test-set adaptation is LFM-A, which achieves significant improvement via adaptation compared to LFM. The comparison to LFM-D (D for descriptive) serves only to show that we can increase the interpretability of the feedback model without suffering performance degradation.
>
> ## Summary
> We sincerely thank the reviewer for taking their time to help us improve this work. We hope we have addressed the reviewer’s concerns and questions. If so, would the reviewer please consider increasing their score to show support for our work?

---

> ### Author Response · Authors · 2024-08-12
>
> Dear Reviewer,
>
> We wanted to send a friendly reminder that we are awaiting your response. With the deadline for the reviewer-author discussion approaching on August 13, we would greatly appreciate it if you could provide feedback at your earliest convenience. We hope we have addressed your concerns and questions. If so, would you please consider increasing your score to show support for our work?
>
> Sincerely,
> Authors

---

> > ### Author Response · Authors · 2024-08-13
> > **Last day of discussion**
> >
> > Dear Reviewer 8DkG,
> > Today is the last day for discussion. Would you please take a look at the author response before the discussion ends? We hope we have addressed your concerns and questions. If so, would you please consider increasing your score to show support for our work?
> > Thank you!
> > Authors

---

### Decision · Program_Chairs · 2024-09-25

**Decision:**

Accept (poster)

**Comment:**

Reviewers' concerns focus on evaluation, particularly limited comparison with baselines and ablations, and on potential limitations of the approach. These type of concerns are common in this emerging field and it's good to see the community push for such improvements. At the same time, it does speak to the innovative nature of work in this field, including this paper, and the authors do satisfactorily address the concerns. What remains is a subjective assessment of significance, which I find sufficient, thus leaning to accept.